# Mutational Effect of Some Major COVID-19 Variants on Binding of the S Protein to ACE2

**DOI:** 10.3390/biom12040572

**Published:** 2022-04-13

**Authors:** Zhendong Li, John Z. H. Zhang

**Affiliations:** 1Shanghai Engineering Research Center of Molecular Therapeutics and New Drug Development, Shanghai Key Laboratory of Green Chemistry & Chemical Process, School of Chemistry and Molecular Engineering, East China Normal University, Shanghai 200062, China; zhendongli1995@126.com; 2CAS Key Laboratory of Quantitative Engineering Biology, Shenzhen Institute of Synthetic Biology, Shenzhen Institute of Advanced Technology, Chinese Academy of Sciences, Shenzhen 518055, China; 3Faculty of Synthetic Biology, Shenzhen Institute of Advanced Technology, Chinese Academy of Sciences, Shenzhen 518055, China; 4NYU-ECNU Center for Computational Chemistry at New York University Shanghai, Shanghai 200062, China; 5Department of Chemistry, New York University, New York, NY 10003, USA; 6Collaborative Innovation Center of Extreme Optics, Shanxi University, Taiyuan 030006, China

**Keywords:** SARS-CoV-2, SARS-CoV-2 variants, spike protein, protein–protein interaction, ASGB, binding free energy, alanine scanning

## Abstract

COVID-19 is caused by severe acute respiratory syndrome coronavirus 2 (SARS-CoV-2), which has many variants that accelerated the spread of the virus. In this study, we investigated the quantitative effect of some major mutants of the spike protein of SARS-CoV-2 binding to the human angiotensin-converting enzyme 2 (ACE2). These mutations are directly related to the Variant of Concern (VOC) including Alpha, Beta, Gamma, Delta and Omicron. Our calculations show that five major mutations (N501Y, E484K, L452R, T478K and K417N), first reported in Alpha, Beta, Gamma and Delta variants, all increase the binding of the S protein to ACE2 (except K417N), consistent with the experimental findings. We also studied an additional eight mutations of the Omicron variant that are located on the interface of the receptor binding domain (RDB) and have not been reported in other VOCs. Our study showed that most of these mutations (except Y505H and G446S) enhance the binding of the S protein to ACE2. The computational predictions helped explain why the Omicron variant quickly became dominant worldwide. Finally, comparison of several different computational methods for binding free energy calculation of these mutants was made. The alanine scanning method used in the current calculation helped to elucidate the residue-specific interactions responsible for the enhanced binding affinities of the mutants. The results show that the ASGB (alanine scanning with generalized Born) method is an efficient and reliable method for these binding free energy calculations due to mutations.

## 1. Introduction

The severe acute respiratory syndrome coronavirus 2 (SARS-CoV-2) infects the human body by binding to the human angiotensin-converting enzyme 2 (ACE2) with its receptor-binding domain (RBD) of the spike protein [1,2,3,4,5,6]. Thus, the interface residues play key roles in the spike protein’s binding to ACE2. At present, the most severe cause of infection is from some Variant of Concern (VOC): Alpha, Beta, Gamma, Delta and Omicron [7,8,9,10,11,12,13,14,15,16]. Especially, the Omicron variant is currently a dominant cause of the resurgence of infection in many countries [14]. These variants are reported to contribute to increased transmission of the virus with some mutations on their S protein [11,17,18,19,20].

The Alpha variant was found in the UK in September 2020, the Beta variant was found in North Africa in May 2020 and Gamma variant was found in Brazil [7,8,9,11]. As shown in Table 1, these three variants have one point in common—that is, the N501Y mutation, which was reported to accelerate the spread of virus [7,20] through stronger binding with ACE2. As for E484K mutation, both Beta and Gamma variants contain it, but the Alpha variant does not [11]. The K417N was found in the Beta variant [11]. The Delta variant received a lot of attention, and the L452R and T478K mutations are considered to be one of the main reasons for the accelerating spread of the virus [19,21,22,23].

Computational studies of mutations of these VOCs have been reported by many groups [24,25,26,27,28]. This paper focuses on these single residue mutations in these VOCs. We first focus on five specific mutations (N501Y, E484K, L452R, T478K and K417N) that were found in Alpha, Beta Gamma and Delta variants [4,7,8,9,10,11,17,18,19,20,21,29], and these mutations are all located on the RBDs of the SARS-CoV-2/ACE2 complex. These five residue mutations have experimental data, so we first analyze these five mutations. In order to analyze the difference in binding affinity between the wild type and the mutants, we performed MD simulations and calculated binding free energies for these complexes based on the ASGB method as well as several different methods. The ASGB method was developed in recent years and has been applied to calculate binding free energies in protein–ligand [30,31] and protein–protein interactions [32,33,34,35,36,37,38]. In this work, we calculated average results from five trajectories for each complex to reduce standard errors from MD simulations. Our calculation results are generally consistent with the available experimental results. We elucidate the molecular mechanism of enhanced binding affinity based on structural information and the contribution of specific residues. For comparison, we also used the MM/GBSA, ASGB-T, TI (thermodynamics interaction) and machine learning methods to calculate the differences in binding free energies between the wild type and the mutants. Recently, a new VOC named Omicron has been reported, which quickly spread to many countries.

The Omicron variant contains 11 mutations in the RBD of SARS-CoV-2. We calculated the ΔΔ*G* values with respect to these mutations. Since no experimental values of binding affinity are available for these Omicron mutations, our result serves as the theoretical prediction.

## 2. Theoretical Method

### 2.1. Molecular Dynamics Simulations

The wild-type X-ray crystal structures of the SARS-CoV-2 RBD/ACE2 (pdbid: 6M0J) [3] were downloaded from the Protein Data Bank (PDB) [15]. The structures of mutated S-protein RBDs that bind with ACE2 are not reported in the experiment, and we built these complex structures based on the wild-type crystal structure (pdbid: 6M0J) by using Rosetta [39]. We used pmemd.cuda in Amber18 [40] with the ff14SB force field [41] for MD simulations with TIP3P-explicit water molecules, and the truncated octahedron box was 12 Å away from the solute atoms. Chloride ions were added to neutralize the system at standard salt concentration (150 mM NaCl). The initial system was minimized for 2000 steps and then heated to 300 K in 200 ps with the protein backbone constrained with 10 kcal·mol^−1^·Å^−2^. In the production stage, we performed five 40 ns long simulations in the NPT ensemble with 2 fs time-step for each system. A total of 200 snapshots were extracted at an interval of 100 ps from the 20 ns to 40 ns trajectory for calculating. Temperature and pressure were controlled using a Langevin thermostat with a collision frequency of 1.0 ps^−1^ and a Berendsen barostat with a pressure relaxation time of 1.0 ps. The particle mesh Ewald (PME) [42] method was used to treat the long-range electrostatic interactions. The non-bonded interactions were truncated with a 12 Å cutoff. The SHAKE [43] algorithm was used to constrain the bonds involving hydrogen atoms. A total of five MD trajectories were used to calculate binding free energies for each system.

### 2.2. Hotspot Prediction by ASGB Method

In our present ASGB method, residues within the 8 Å range of the protein–protein interaction interface were mutated into alanine, and the result was calculated using trajectories extracted between 20 and 40 ns. This method uses alanine scanning with MM/GBSA to calculate protein–ligand and protein–protein interactions by quantitatively analyzing the contribution of individual residues. The dielectric constants 1, 3, 5 were used for nonpolar, polar and charged residues, respectively [30,33,34,37,44]. We calculated the binding free energy difference between the wild type (*x* amino acid) and the mutant (*a* amino acid) in protein *A* by the following equation [30,32,36,37,45,46]:(1)$$\Delta \Delta G_{bind}^{A^{x}\to A^{a}}=\Delta \Delta G_{gas}^{A^{x}\to A^{a}}+\Delta \Delta G_{sol}^{A^{x}\to A^{a}}$$
where
(2)$$\Delta \Delta G_{gas}^{A^{x}\to A^{a}}\approx \Delta \Delta H_{gas}^{A^{x}\to A^{a}}\approx \Delta \Delta E_{vdw}^{A^{x}-B}+\Delta \Delta E_{ele}^{A^{x}-B}$$

The $\Delta E_{vdw}^{A^{x}-B}$ and $\Delta E_{ele}^{A^{x}-B}$ are, respectively, the differences in van der Walls interaction and electrostatic interaction energies between the alanine mutant and the wild type in protein *B*. We used the MM/GBSA method to calculate the solvation component in Equation (1):(3)$$\Delta \Delta G_{sol}^{A^{x}\to A^{a}}=(G_{sol}^{A_{}^{a}B}-G_{sol}^{A_{}^{x}B})-(G_{sol}^{A_{}^{a}}-G_{sol}^{A_{}^{x}})$$
where $\Delta G_{sol}^{A^{a}B}$ $\Delta G_{sol}^{A^{x}B}$$\Delta G_{sol}^{A^{a}}$ and $\Delta G_{sol}^{A^{x}}$ are the solvation free energies of the protein–protein complex *A^x^B*, protein–protein complex *A^a^B*, protein *A^x^* and protein *A^a^*, respectively. In this paper, the protein *A* and protein *B* are the S protein and ACE2, respectively, and the protein *A^x^* and protein *A^a^* represent the S protein before and after the alanine scanning, respectively. The solvation free energy ($\Delta G_{sol}$) was calculated by:(4)$$\Delta G_{sol}=\Delta G_{gb}+\Delta G_{np}$$
where $G_{gb}$ and $G_{np}$ are the electrostatic solvation free energy and the nonpolar solvation free energy, respectively. $G_{np}$ is given by an empirical solvent-accessible surface area (SASA) formula:(5)$$G_{np}=\gamma \mathrm{SASA}+\beta$$

The *γ* and *β* values we used here were the standard values of 0.00542 kcal/(mol·Å^2^) and 0.92 kcal/mol.

For a general mutation from amino acid *x* to *y* in protein *A*, we calculated the difference in binding free energy between two alanine mutations by the relation:(6)$$\Delta \Delta G_{bind}^{A^{x}\to A^{y}}\approx \Delta \Delta G_{bind}^{A^{y}\to A^{a}}-\Delta \Delta G_{bind}^{A^{x}\to A^{a}}$$
where $\Delta \Delta G_{bind}^{A^{x}\to A^{a}}$ or $\Delta \Delta G_{bind}^{A^{y}\to A^{a}}$ is obtained by Equations (1)–(3). In this work, we mainly discuss the effect of mutated residues in S protein, so the *A^x^* and *A^y^* are the wild type and the mutants of S protein, respectively.

## 3. Results and Discussion

### 3.1. ASGB Analysis for Single Point Mutations in Alpha, Beta, Gamma and Delta

Here, we performed independent MD simulations for different systems and calculated binding free energy contributions of the mutated residues respectively by using the ASGB method. The calculated results of binding affinity for interface residues on the wild type and five mutants are shown in Figure 1. The results were obtained by averaging over five independent trajectories of each system. We can note that the specific contributions of N501Y, E484K, L452R and T478K are higher than that of wild type, and the specific contribution of K417N is lower than that of wild type. The calculated results are consistent with experimental results. Here, we analyze each mutation according to our calculation results and structural information.

#### 3.1.1. N501Y

The N501Y mutation exists in Alpha, Beta and Gamma variants. As shown in Figure 1a, the binding contribution of N501Y residue (7.7 kcal/mol) is significantly higher than N501 (3.4 kcal/mol). In Table 2, the computational ΔΔ*G* of N501Y is 4.3 kcal/mol, in which the van der Waals interaction contributed to 3.8 kcal/mol and that is the main reason for the large binding free energy calculated. According to the local structure in Figure 2, a π–π interaction was formed between ACE2 residue Y41 and N501Y but not with N501. We consider that the π–π interaction between Y501 and Y41 is the main reason of the enhanced binding affinity. Although the calculated value is larger than the experimental data, we believe that such overestimation is related to the overestimation of the van der Waals interaction energy.

#### 3.1.2. E484K

The E484K mutation is also reported to enhance the binding affinity of the S protein to ACE2, and this mutation exists in Beta and Gamma variants. Our calculation result also shows that the E484K mutation enhances the binding. We notice that the energy contribution of E484 is −2.2 kcal/mol. and that of E484K is 1.8 kcal/mol., as shown in Figure 1b. In Table 2, the ΔΔ*E_ele_* and ΔΔ*E_gb_* change in E484K mutation and resulted in an increased binding affinity. The change in electrostatic interaction is mainly due to the fact that electrostatic repulsion exists between E484 and E35 of ACE2 in the WT, while the mutant E484K creates electrostatic attraction with E35 (shown in Figure 2).

#### 3.1.3. L452R

The Delta variant spread rapidly all over the world. The L452R mutation was found in Delta variants, which is considered to be one of the reasons for the accelerating spread of variants. In Figure 1c, the calculated binding free energy of L452R (2.4 kcal/mol) is larger than the wild type. In order to explore the specific reasons for this difference, we show the detailed data of binding free energy in Table 2. The change in the electrostatic interaction (ΔΔ*E_ele_* + ΔΔ*E_gb_*) is the main reason for the change in the binding free energy contribution of L452(R) because the positively charged L452R and negatively charged E35 of ACE2 produce electrostatic attraction, while the WT L452 does not (Figure 2).

#### 3.1.4. T478K

The Delta variant also contains a T478K mutation, and the van der Waals interaction is the main reason for the change in binding affinity (in Figure 1 and Table 1). Although lysine is a positive residue, there is no negative residue on ACE2 that closely interacts with T478K (Figure 2), so the electrostatic interaction (ΔΔ*E_ele_* + ΔΔ*E_gb_*) changes little. The van der Waals interaction plays a key role in increasing binding free energy; the van der Waals interaction of T478K is 0.9 kcal/mol. larger than that of T478.

#### 3.1.5. K417N

The K417N mutation was found in the Beta variant, and it is thought to cause immune escape [47]. In Table 2, the decrease in electrostatic interaction (ΔΔ*E_ele_* + ΔΔ*E_gb_*) is the main reason for the decrease in binding free energy of the K417N complex. Our calculation results were supported by structural information. We note that the mutant K417N destroyed the salt bridge that exists between ACE2 residue D30 and K417 (shown in Figure 2). We believe that this is the reason for the weakening of binding affinity in K417N system.

### 3.2. Comparison of Results Using Different Methods

In order to analyze these mutations more systematically, we used different methods (MM/GBSA, ASGB-T, TI and Mutabind2) to analyze the differences in the binding free energy of these mutations.

#### 3.2.1. MM/GBSA Method

We calculated wild-type and mutants binding free energy by the MM/GBSA method [48,49,50]. A total of 200 snapshots were selected from final 20 ns trajectories at an interval of 100 ps, and the calculated complex binding free energy is the average of five independent trajectories (shown in Table 3). According to Figure 3, the results of the MM/GBSA method are generally overestimates of the ASGB method. Compared with MM/GBSA method, the data of the ASGB method are closer to experimental results. Although the MM/GBSA method can obtain qualitative calculation results of protein–protein interaction binding free energy, the quantitative results are not satisfactory. In this view, the ASGB method is more suitable than MM/GBSA method.

#### 3.2.2. ASGB-T Method

In the ASGB method used in this paper, only the effect of a single point mutation from the WT to alanine is evaluated. In the ASGB-T (ASGB-Total) method [30,33,34,37], we calculate their complex binding affinity by summing over binding free energy contributions of the interface residues, which are shown in Figure 1. Compared with the ASGB method, the ASGB-T method takes into account the overall impact of residue mutations:(7)$$\Delta \Delta G_{bind}^{A^{x}\to A^{y}}\approx \sum \Delta \Delta G_{bind}^{A^{y}\to a}-\sum \Delta \Delta G_{bind}^{A^{x}\to a}$$

By using the ASGB-T method, the complex binding free energy of mutants is higher than that of the wild type, and these results are averaged from five independent trajectories for each complex (shown in Table 3). We can note that among these mutants, the binding free energy of the n501y complex is the highest (4.4 kcal/mol). In the experimental results, the contribution of n501y mutation is also the largest among the five mutations. The E484K mutation was found in Beta and Gamma variants with higher binding affinity than wild type. In our calculation, the E484K increased the binding free energy contribution with 3.7 kcal/mol, which is overestimated relative to the experimental data. The calculated ΔΔ*G* of the L478R mutation is 3.0 kcal/mol, the T478K’s is 2.2 kcal/mol and K417N’s is −4.2 kcal/mol. In Figure 3 and Table 3, we can note that our calculated value of these five mutations are all overestimated by the ASGB-T method, mainly due to the overestimation of the van der Waals interaction. Compared with the ASGB-T method, the calculated data of ASGB are closer to the experimental results.

#### 3.2.3. TI Method

In addition, the thermodynamic integration method was also used in calculating the binding free energy of wild type and mutants. The preparation and equilibration steps of mutated structures are similar to the above MD simulations with the AMBER-TI model [51,52,53,54]. We set a total of 11 λ windows range 0.0 to 1.0 with an interval of 0.1. For each window, we ran a 1 ns simulation by using pmemd (CPU) for the equilibration of the complex. After the CPU equilibration, we performed 20 ns simulations for each window by using pmemdGTI (GPU). The final result Δ*G* was obtained based on the TI gradient (dV/dL) integration from the last 15 ns simulations by using the script from the Amber tutorial (http://ambermd.org/tutorials/advanced/tutorial9/index.html#analysis (accessed on 8 March 2022)) (shown in Table 3). In our TI calculation, the contribution of N501Y (3.1 kcal/mol) is the largest among all mutants that is consistent with experimental results. The binding free energy values of E484K (1.9 kcal/mol), L452R (2.4 kcal/mol) and T478K (2.3 kcal/mol) are not different, and the experimental values corresponding to these mutations are also very similar. Of course, our calculations are often overestimated. The binding affinity of the K417N mutation (−1.8 kcal/mol) was close to the experimental data (−0.8 kcal/mol). In addition, although the calculation accuracy of the ASGB method is slightly worse than that of TI method (as shown in Figure 3), the calculation cost of the TI method is one order of magnitude higher than that of ASGB method.

#### 3.2.4. Mutabind Method

Recently, there are many works that have been reported for predicting the binding free energy effects of residues mutation in protein–protein interaction by a machine learning method, such as PPI, FoldX and Mutabind. In this work, we predicted the effects of these five mutations by using Mutabind2, which was published in February 2020. We submitted the relevant information on the website (https://lilab.jysw.suda.edu.cn/research/mutabind2/ (accessed on 8 March 2022)) and obtained the computational results (in Table 3). For N501Y, E484K and T478K, the results of Mutabind2 are contrary to the experimental phenomenon. Machine learning methods may not be effective in dealing with untrained problems. Due to the limitations of machine learning methods, the method based on a physical model may be more suitable for this work.

We also performed the scatter diagram of experimental and calculated values of various methods and the correlation between them (shown in Figure 4). We note that the correlation value of the TI method is the highest among these methods and that the Mutabind2 method is the worst, failing to give the correct prediction of direction. The correlations of ASGB, ASGB-T and GBSA methods are quite similar, with the GBSA method overestimating the relative binding affinity most.

### 3.3. ASGB Analysis for Single Point Mutations in Omicron Variants

The SARS-CoV-2 Omicron variant has emerged in many countries and contains 11 mutations on the interface of SARS-CoV-2 RBD/ACE2 [14,16]. According to the reported experimental data, the Omicron variant binds to ACE2 with enhanced affinity relative to wild type.

Among these 11 mutations, 8 mutations have not been reported in the previous variants of interest (VOCs). We used the ASGB method to calculate the effect of these eight mutations on the binding free energy of the complex. The calculation results of the binding affinity of interface residues on wild type and eight mutants are shown in Figure 5. Here, we analyze each mutation according to the calculation results and structural information.

#### 3.3.1. Y505H

The calculated ΔΔ*G* of Y505H is −2.2 kcal/mol (Figure 5a), and the decreased van der Waals interaction is −1.5 kcal/mol (shown in Table 4). The aromatic side chains of Y505 participate in the van der Waals interaction with E37 and R393, but Y505 was mutated into Y505H (Figure 6), which decreased the van der Waals interaction with E37 and R393.

#### 3.3.2. Q498R

According to Figure 5b and Table 4, the binding free energy contribution of Q498R (8.9 kcal/mol) is larger than Q498′s (5.1 kcal/mol), and the electrostatic interaction of Q498R is stronger than that of the Q498 system. This view is also revealed in the salt bridge between D38 and Q498R is not available in the Q498 system (shown in Figure 6).

#### 3.3.3. Q493K

The situation of Q493K is similar to that of Q498R, in that the salt bridge between Q493K and E35 are not existent in the wild-type complex (shown in Figure 6). The difference in electrostatic interaction (ΔΔ*E_ele_* + ΔΔ*E_gb_*) is shown in Table 4, which increased the binding free energy of the complex. We can also note that the difference in van der Waals interaction is −2.1 kcal/mol, which is decreased after Q493K mutation, while the strong electrostatic interaction caused the Q493K and E35 to get too close, decreasing the van der Waals interaction.

#### 3.3.4. E484A

The E484A mutation increased the complex binding free energy for 2.3 kcal/mol in our calculated data (shown in Figure 5d and Table 4). In the wild-type complex, the E484 repelled the E35 by electrostatic repulsion, and the higher affinity is due to the disappearance of electrostatic repulsion between E35 and E484 (Figure 6).

#### 3.3.5. G496S

The van der Waals interactions from the G496 provide lower binding affinity (shown in Figure 5e and Table 4), and G496 was mutated into G496S (shown in Figure 6), which increased the van der Waals interaction with ACE2.

#### 3.3.6. S477N

The contribution of the S477N mutation is larger than that of the wild-type S477 by 1.1 kcal/mol (shown in Figure 5f), and the main cause is the difference in the van der Waals interaction (Table 4). This result is consistent with the experiment result, and the change in the van der Waals interaction is mainly because S477N contains a longer side chain than S477 (shown in Figure 6).

#### 3.3.7. G446S

The binding free energy of both G446 and G446S contributes almost little to the SARS-CoV-2/ACE2 complex (Figure 5g and Table 4). The mutated structure situation of G446S is similar to G496S’s (shown in Figure 6), and the differences in their binding free energy contribution are not significant.

#### 3.3.8. N440K

Compared with wild-type system, the residue of N440 was mutated to N440K, which interacted with E329 (shown in Figure 6), resulting in the stronger electrostatic interaction between N440K and E329 (Table 4). We think that this is the reason for the increased contribution of the N440K system.

These mutations (Y505H, Q498R, Q493K, E484A, G496S, S477N, G446S and N440K) have not been reported in previous Variants of Concern (VOCs). We performed MD simulations and calculated the ΔΔ*G* of these mutations by the ASGB method and demonstrated the influences of these mutations based on our calculated data at the molecular mechanism level. We predicted that mutations Q498R, Q493K, E484A, G496S, S477N and N440K would enhance the binding affinity.

Many effective neutralizing antibodies have been found to bind to RBD. Here, we analyzed a total of 13 mutations on the RBD interface. These mutations were found in five VOCs, which may affect the binding of antibodies to the S protein, thus affecting the effectiveness of the vaccines. We analyzed the effect of these mutations on the binding free energy of the SARS-CoV-2/ACE2 complex by theoretical calculation, which is helpful in the design of new antibodies.

## 4. Conclusions

In this work, we performed MD simulations and calculated the binding free energies of the S protein to ACE2 for some major mutations (a total of 13 mutations) present in the RBD interface region of the S protein in Alpha, Beta, Gamma, Delta and Omicron variants. Our calculations show that these mutations can enhance the binding affinity of the complex, except for K417N and Y505H mutations. The calculations provide service for new antibody design and also provide insight on the molecular mechanism of these mutations, while revealing the detailed residue-specific interactions that are responsible for the mutational effect. Moreover, we compared several methods for calculating the mutational effect of SARS-CoV-2 on binding to ACE2. The TI method is more accurate, but it is also the most expensive, while the ML method gives the wrong predictions. On the other hand, the ASGB method provides a good balance in terms of accuracy and efficiency.

## Figures and Tables

**Figure 1 biomolecules-12-00572-f001:**
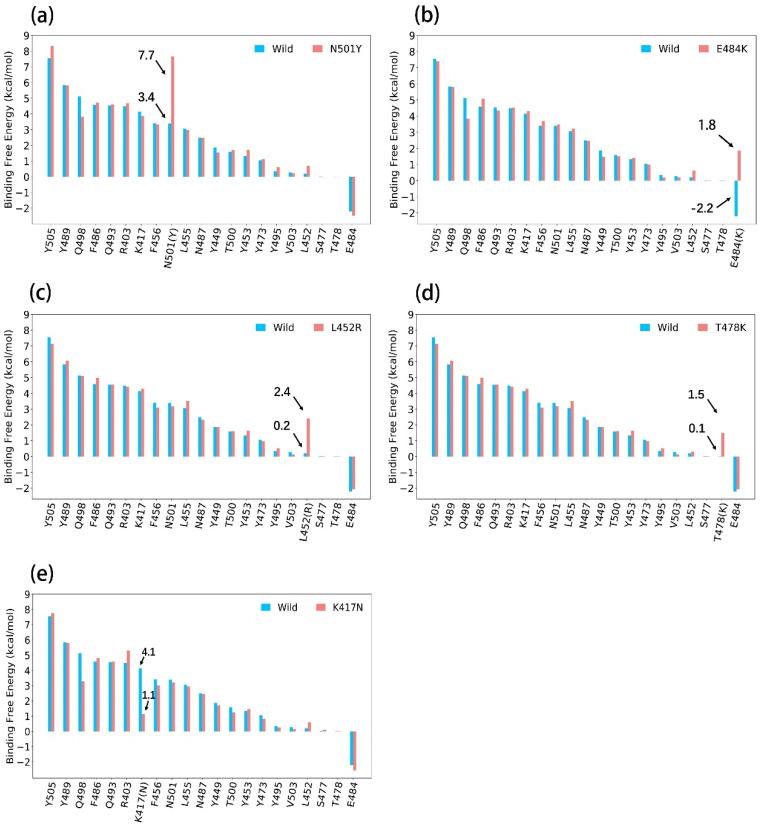
Residue-specific binding free energy data of the S protein between the wild type and the different mutants in binding to ACE2 from the ASGB calculation. (Calculated results were averaged over 5 trajectories separately). The (**a**–**e**) show the specific binding affinity contributions of interface residues on S protein for wild type and N501Y, E484K, L452R, T478K and K417N complex.

**Figure 2 biomolecules-12-00572-f002:**
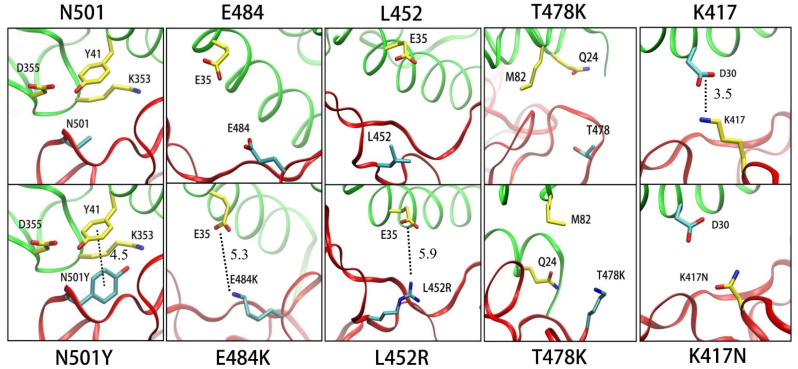
The local interaction structures of the wild type (above figures) and mutants (below figures) in SARS-CoV-2 S-protein/ACE2 complexes. The wild-type N501 and mutated N501Y where the π–π interaction is indicated by the dashed line between Y41 and N501Y. The wild type E484 and mutated E484K. The wild-type L452 and mutated L452R. The wild-type T478 and mutated T478K. The wild-type K417 and mutated K417N.

**Figure 3 biomolecules-12-00572-f003:**
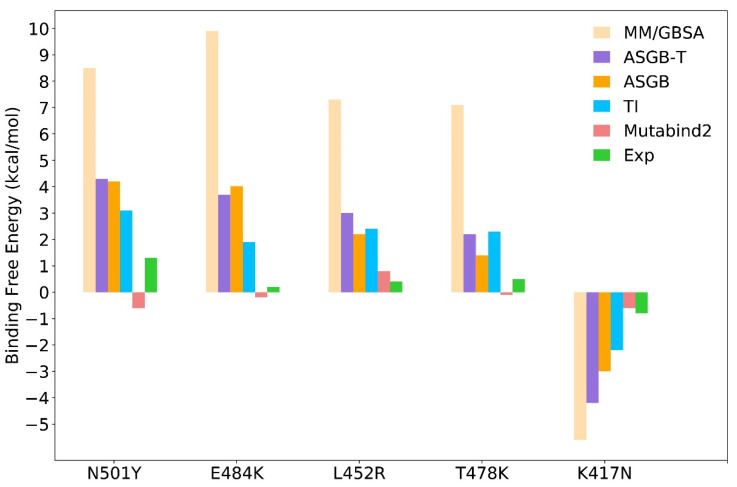
Computational results of these five mutations (N501Y, E484K, L452R, T478K and K417N) by using different methods. The yellow, purple, orange, blue, red and green bars represent the MM/GBSA method, ASGB-T method, ASGB method, TI method, Mutabind2 method and experimental results, respectively.

**Figure 4 biomolecules-12-00572-f004:**
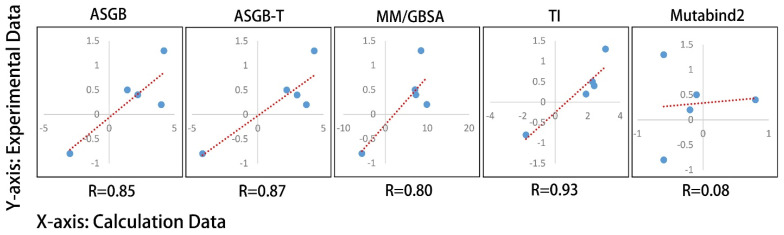
Scatter diagram of experimental and calculated values of various methods. The red dotted line is the fitting line between calculated and experimental values. The R values represent the linear correlation coefficient between the experimental and calculated values of different methods.

**Figure 5 biomolecules-12-00572-f005:**
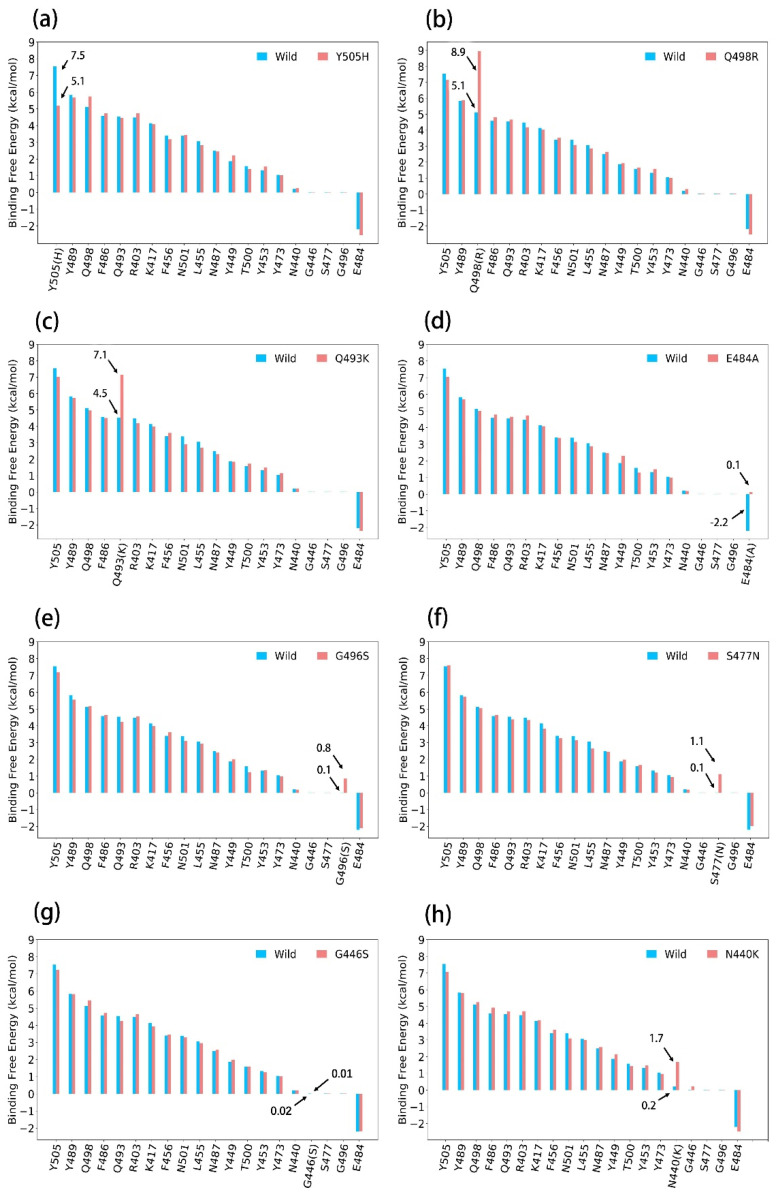
Residue-specific binding free energy data of the S protein between the wild type and the different mutants in binding to ACE2 from the ASGB calculation. (Calculated results were averaged over 5 trajectories separately). (**a**–**h**) show the specific binding affinity contributions of interface residues on S protein for wild type and Y505H, Q498R, Q493K, E484A, G496S, S477N, G446S and N440K complex.

**Figure 6 biomolecules-12-00572-f006:**
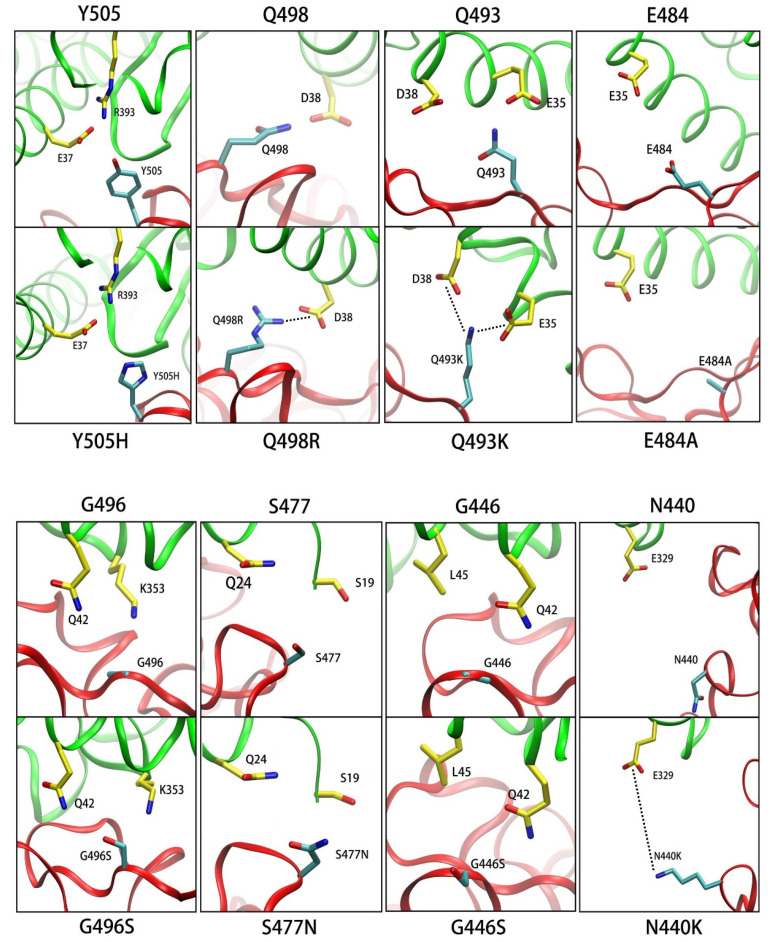
The local interaction structures of the wild type (above figures) and mutants (below figures) in SARS-CoV-2 S-protein/ACE2 complexes. The wild-type Y505 and mutated Y505H. The wild-type Q498 and mutated Q498R. The wild-type Q493 and mutated Q493K. The wild-type E484 and mutated E484A. The wild-type G496 and mutated G496S. The wild-type S477 and mutated S477N. The wild-type G446 and mutated G446S. The wild-type N440 and mutated N440K.

**Table 1 biomolecules-12-00572-t001:** Major single point mutations in Alpha, Beta, Gamma, Delta and Omicron variants of SARS-CoV-2 in the RBD that are studied in this work.

Mutation Sites	Alpha	Beta	Gamma	Delta	Omicron
K417N		√			√
N440K					√
G446S					√
L452R				√	
S477N					√
T478K				√	√
E484A					√
E484K		√	√		
Q493K					√
G496S					√
Q498R					√
N501Y	√	√	√		√
Y505H					√

**Table 2 biomolecules-12-00572-t002:** Calculated binding energy differences between specific single point mutants and the wild-type SARS-CoV-2 S protein in binding to ACE2 by the ASGB method. The energy values are in kcal/mol.

Mutation	ΔΔ*E_vdw_*	ΔΔ*E_ele_*	ΔΔ*E_gb_*	ΔΔ*E_np_*	ΔΔ*G_cal_*	ΔΔ*G*_exp_ ^a^
N501Y	3.8	0.4	−0.1	0.3	4.2	1.3
E484K	0.1	84.8	−80.7	−0.1	4.0	0.2
L452R	0.3	32.4	−30.6	0.1	2.2	0.4
T478K	0.9	10.1	−9.7	0.1	1.4	0.5
K417N	−0.5	−56.8	54.3	−0.2	−3.0	−0.8

^a^ Experimental values were converted using the relation $\Delta G_{\exp}=-RTInK_{D}$ at T=298K where the $K_{D}$ values for the wild type and mutant are from [11,19,20,21].

**Table 3 biomolecules-12-00572-t003:** Calculated binding energy differences between specific mutations and the wild-type SARS-CoV-2 S protein in binding to ACE2 using different methods.

Systems	ASGB	ASGB-T	MM/GBSA	TI	Mutabind2	ΔΔ*G*_exp_ ^a^
N501Y	4.2	4.3	8.5	3.1	−0.6	1.3
E484K	4.0	3.7	9.9	1.9	−0.2	0.2
L452R	2.2	3.0	7.3	2.4	0.8	0.4
T478K	1.4	2.2	7.1	2.3	−0.1	0.5
K417N	−3.0	−4.2	−5.6	−1.8	−0.6	−0.8

^a^ Experimental values are converted using the relation $\Delta G_{\exp}=-RTInK_{D}$ at T=298K where the $K_{D}$ values for the wild type and mutant are from [11,19,20,21]

**Table 4 biomolecules-12-00572-t004:** Calculated binding energy differences between specific single point mutants (Y505H, Q498R, Q493K, E484A, G496S, S477N, G446S and N440K) and the wild-type SARS-CoV-2 S protein in binding to ACE2 by the ASGB method. The energy values are in kcal/mol.

Mutation	ΔΔ*E_vdw_*	ΔΔ*E_ele_*	ΔΔ*E_gb_*	ΔΔ*E_np_*	ΔΔ*G_cal_*
Y505H ↓	−1.5	−1.0	0.3	−0.2	−2.3
Q498R ↑	2.2	61.2	−60.1	0.5	3.8
Q493K ↑	−2.1	62.8	−58.1	0.1	2.7
E484A ↑	−0.1	−37.1	39.2	−0.1	2.3
G496S ↑	0.7	0.4	−0.5	0.2	0.8
S477N ↑	0.9	0.1	0.2	−0.1	1.1
G446S =	0.1	0.1	−0.2	0	0
N440K ↑	0.5	45.5	−44.7	0.2	1.5

## Data Availability

The data are available upon request.

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
