# Peer review of "Mutational Effect of Some Major COVID-19 Variants on Binding of the S Protein to ACE2"

_biomolecules, 2022, doi:10.3390/biom12040572_

Round 1

Reviewer 1 Report

The paper describes a theoretical analysis of the effect of some major mutants of the spike protein of SARS-CoV-2 in relation to its binding to the human angiotensin-converting enzyme 2 (ACE2). The analysis is well described and reproducible. Also different methods are campared. I would like to read more motivations and practical applications of the results in relation to the design of new vaccines.

Author Response

Please see attached response.

Reviewer 2 Report

The manuscript by Li and Zhang is an interesting analysis of methods to study the effect of amino-acid mutations in the main COVID-19 variants of interest.

The work is presented in a clear way, and indicates a favorable balance between accuracy and speed using the ASGB method.

I am in favour of the publication, pending some corrections and clarifications.

Abstract:

“accelerated the spread” is that so? Are we sure the mutations have actually accelerated the spread? What is the physiological explanation for that?

Introduction:

Table 1 is nice. Would be nice to put an asterisk on the mutations considered in this paper. Referring to, or showing, a tree diagram to show the incidence of mutations will be useful (could be as supplemental material). In the caption or in the main text it needs a justification to limit to these 5 mutations (e.g., are they sufficient to map the differences across the main VOCs?)

“a new VOC named omicron have”, change to “has”

Theoretical method

2.1 The authors cite PDB:6M0J. Are there other PDBs for the VOCs to be cited? It’s not necessary to use them for the calculations, but they should be acknowledged (7WBL, for instance is that of Omicron RBD with ACE2=

Results and Discussion:

3.1 All the first paragraph repeats the methods bit, I believe these should be removed and paragraph should focus on the results only.

Figure 1 (and 5) could show better which is the mutation considered (instead of a, b, c…)

In the description of single mutants it would be good to specify when aminoacid is from ACE2  For instance: under N501Y authors describe Y41 which is from ACE, but this is not clear. This has to be corrected throughout the paper

K417N: “thought to cause immune escape” needs a reference to some relevant paper

Figure 2: new interactions should be shown better. I see the dashed lines but they are faint and could do with fainter backbone ribbons and much thicker dashed lines between the aminoacids involved in the interaction. Also a distance measurement would be meaningful.

MM/GBSA method: Here and elsewhere “compare” should read “compared”

TI method:

it is not true that the binding energies are not different from ASGB. The trend is the same, but E484K, K417N and N501Y seem double to me.

The authors say TI is more computationally intense, albeit more precise. It would be nice to explore how ASGB method, performed on more simulations to match the computational time of those currently used for TI would behave (i.e., does ASGB improve significantly to match TI perfomance with less computational effort?) Also, the values are still way different from the experimental ones. Can the authors comment? I can think of the ratio between experimental and calculated: 18.5 times higher for E484K, 7.5 times for L452R and always over 3 times higher in general.

Figure 4: this shows correlation between experimental and calculated values. The coordinate systems should be indicated. It is clear that, apart from Mutabind2, all others have decent correlations, but reporting R is somewhat misleading… one would expect a good correlation to be R^2 above 0.95 (i.e. R>97). Can this convergence be achieved with any of these methods? If so, at what cost (computational) or compromise (scaling factor)?

Table 4 could show the effect of up/down regulation of each single mutation (e.g. graphically displaying up/dn arrow) or no effect (e.g. with an = sign)

4 Conclusion

“Our calculations show that almost all these mutations” (inversion of words is needed as indicated)

“the ML method Mutabind2”. I believe it is necessary to cite the name of the method as other ML-based ones could give better results (also because ML is heavily skewed towards training set choice, so improving training could lead to better results than those presented here.

Can the authors comment something more on the balance between performance and accuracy? Would ASGB be sufficient to give good predictions? As said above the correlation of R=0.87 (R^2 = 0.73) is something good for economics, but unacceptable for physics (and this is biophysics). How can at least R^2 = 0.9 be achieved? At what expenses?

Author Response

Please see attached responses.

Reviewer 3 Report

The authors of the manuscript, “Mutational effect of some major COVID-19 variants on the binding of the S protein to ACE2” have used 40 ns MD simulation and different binding free energy calculations along with residue energy decomposition on the different SARS-CoV-2 variants to establish the role of interacting residues in the viral protein with the hACE2. The study is interesting, but in my opinion, it repeats previously published work.

Some published work~

Inferring the stabilization effects of SARS-CoV-2 variants on the binding with ACE2 receptor; 10.1038/s42003-021-02946-w.

Molecular insights into receptor binding energetics and neutralization of SARS-CoV-2 variants: 0.1038/s41467-021-27325-1

Key Interacting Residues between RBD of SARS-CoV-2 and ACE2 Receptor: Combination of Molecular Dynamics Simulation and Density Functional Calculation: /10.1021/acs.jcim.1c00560

Mutations on RBD of SARS-CoV-2 Omicron variant result in stronger binding to human ACE2 receptor: 10.1016/j.bbrc.2021.12.079

Scanning the RBD-ACE2 molecular interactions in Omicron variant: 10.1016/j.bbrc.2022.01.006

Additionally, more stringent analysis has been performed in published work than in the present manuscript. Keep the published work into consideration, in my opinion, the present study lacks significant data for publication.

Author Response

Please see attached response.

Round 2

Reviewer 3 Report

Thanks to the authors for adding the clarifications and corrections in the revised manuscript. In my opinion, with updated references and other corrections, the manuscript can be considered for publication.